# Learnings from Separate *Aconitum* Poisonings in British Columbia and Ontario, Canada in 2022

**DOI:** 10.3390/toxins17030125

**Published:** 2025-03-07

**Authors:** Lorraine McIntyre, Stefanie Georgopoulos, Dorianna Simone, Emily Newhouse, JoAnne Fernandes, David A. McVea, Arnold Fok, Ania-Maria McIntyre, Bryn Shurmer, Marie-Claude Gagnon, Michael Chan, Marina Chiaravalloti, Nikita Saha Turna, Debra Kent, Dennis Leong, Katherine Paphitis, Christina Lee

**Affiliations:** 1Environmental Health Services, BC Centre for Disease Control, Vancouver, BC V5Z 4R4, Canada; lorraine.mcintyre@bccdc.ca (L.M.); nikita.sahaturna@bccdc.ca (N.S.T.); 2Office of the Chief Medical Officer of Health, Ontario Ministry of Health, Toronto, ON M5G 2C8, Canada; stefanie.georgopoulos@ontario.ca; 3Regional Municipality of York, Public Health, Newmarket, ON L3Y 2R2, Canada; dorianna.simone@york.ca (D.S.); joanne.fernandes@york.ca (J.F.); 4Fraser Health Authority, Surrey, BC V3T 5X3, Canada; emily.newhouse@fraserhealth.ca (E.N.); arnold.fok@fraserhealth.ca (A.F.); 5Office of Food Safety and Recall, Canadian Food Inspection Agency, Ottawa, ON K1A 0Y9, Canada; ania-maria.mcintyre@inspection.gc.ca; 6Saskatoon Toxin Laboratory, Canadian Food Inspection Agency, Saskatoon, SK S7N 2R3, Canada; bryn.shurmer@inspection.gc.ca; 7Genotyping/Botany Laboratory, Canadian Food Inspection Agency, Ottawa, ON K2J 4S1, Canada; marie-claude.gagnon@inspection.gc.ca; 8Natural Health and Food Products Research Group, British Columbia Institute of Technology, Burnaby, BC V5G 3H2, Canada; michael_chan@bcit.ca; 9Centre of Forensic Sciences, Ministry of the Solicitor General, Toronto, ON M3M 0B1, Canada; marina.chiaravalloti@ontario.ca; 10BC Drug and Poison Information Centre, Vancouver, BC V5Z 4R4, Canada; kent@dpic.ca (D.K.); leong@dpic.ca (D.L.); 11Public Health Ontario, Toronto, ON M5G 1M1, Canada; christina.lee@oahpp.ca

**Keywords:** aconitine, poisoning, monkshood, outbreak, toxin, mislabeling, adulteration, misrepresentation

## Abstract

Background: Three aconitine poisoning events occurred in two Canadian provinces in 2022: one in British Columbia (BC) and two in Ontario (ON). Aconitine is a potent alkaloid found in several species of the plant *Aconitum*, containing cardiotoxins and neurotoxins. It is used in traditional Chinese medicine (TCM) for pain management, and in powdered form, *Aconitum* is similar in appearance to sand ginger (*Kaempferia galanga*), which can lead to poisonings from misidentification and mislabeling. Methods: Aconitine poisoning is rare in Canada; here, we compare communications, collaborations, laboratory testing options and actions during investigations. Results: Fourteen cases occurred from the consumption of sand ginger: in BC (*n* = 2), purchased at an Asian health food store; in ON (*n* = 11), *Kaempferia galanga* powder (KGP) spices were used to prepare meals at a restaurant, and in one ON case, KGP was purchased. Traceback found product imported from China contained aconitine levels ranging from 1304 to 5500 ppm. Later investigations revealed mislabeling of *Aconitum* as KGP from the same imported lot (January 2020). Plant DNA testing found no KGP in any spice packets, including lots not linked to illness, suggestive of adulteration. Conclusion: Method development for aconitine in BC led to an improved response time for testing in ON. BC and ON updated outbreak response protocols and communications.

## 1. Introduction

*Aconitum* or aconite is a herbaceous plant within the Ranuculaceae (buttercup) family and native to the Northern Hemisphere, including parts of North America, Europe and Asia [1]. It includes the species commonly known as monkshood (*Aconitum nupellus*), so named from the blue flower resembling the hood of a monk’s cloak, the species known as wolfsbane (*A. luparia*) and devil’s helmet, with over 300 species in the genus [1].

Raw aconite leaves and roots are poisonous, containing alkaloids in high concentrations [1,2]. Alkaloids of importance include aconitine, mesaconitine and hypaconitine, which are neurotoxic and cardiotoxic, capable of causing severe and fatal cardiac damage, liver and kidney toxicity, and neurotoxicity [2,3]. Symptoms of toxicity typically begin within 30 min of ingestion, ranging from 3 min to 2 h [1,3]. Symptoms may vary depending on the dose ingested. Cardiac symptoms may include hypotension, cardiotoxicity and arrhythmias including bradycardia, tachycardia and fibrillation [1,3,4]. In addition to cardiac symptoms, exposed cases may present with gastrointestinal symptoms of vomiting, nausea, abdominal pain and diarrhoea, and neurological issues of paresthesias of facial numbness and muscle numbness and weakness [1,4]. Treatment is supportive and there is no specific antidote.

Despite toxicity, *Aconitum* plants (roots, stems, leaves, flowers) have been widely used in traditional Chinese medicine (TCM) for over 2000 years as analgesic, anti-inflammatory, anti-cancer, anti-viral and cardiotonic agents [5]. In TCM and for homeopathic preparations of herbal soups and meals, *Aconitum* roots (named “fuzi”, “caowu” and “chuanwu”) and leaves are processed by soaking in water, boiling, steaming and frying to reduce the toxic alkaloid content [4,6,7,8]. Heating hydrolyzes the diester alkaloids to the less toxic constituents of aconine, mesaconine and hypaconine and is an effective way of reducing toxicity [3,4]. Tinctures are prepared by adding unprocessed *Aconitum* into alcohol, which is taken orally as drops [4]. Because alkaloids dissolve well in alcohol, these tinctures can contain high alkaloid levels [4]. Ingestion of raw aconite without appropriate processing to reduce the toxic alkaloid content can result in intoxication. Aconitine poisoning events are rare in North America, and they typically occur due to the herb being mistaken for another herb, or due to accidental ingestion by children.

There is a low safety margin between a therapeutic and a toxic dose. A typical lethal dose of ingested aconitine in an adult is 5 mg, and a minimum lethal dose from oral administration of pure aconitine is estimated to be as low as 1–2 mg, with severe cardiac arrhythmias observed following ingestion of 2 mg [9,10] and toxic effects reported after ingestion of only 0.2 mg of aconitine [11].

Aconitine poisoning is frequently reported in China, linked to the consumption of non-toxic herbs when products are contaminated with *Aconitum* materials [12]. In North America, reports are less frequent, but examples include two fatalities that occurred from the consumption of medicinal tea purchased from a herbal shop in San Francisco in 2018 [13]. Adulteration and contamination of herbal medicines occur through plant misidentification and mislabeling, often going unreported unless adverse reactions prompt an investigation [13,14,15,16]. In 2022, food prepared with sand ginger spice containing *Aconitum* poisoned multiple people in British Columbia (BC) and Ontario (ON). Regional, provincial and federal public health agencies investigated and responded to these poisonings. Here, we report the shared experiences and learnings when dealing with these unusual toxic poisonings, comparing how cases were identified, inspection approaches and communication networks between the separate investigations.

## 2. Results

A description of all partners involved in the investigations is provided in Appendix A and an overview of the BC and ON investigations is provided in Appendix A.

### 2.1. Case Descriptions

In total, 14 aconitine poisoning cases occurred in BC (n = 2) and ON (n = 12) in 2022 linked to an imported spice from China. All cases reported the onset of symptoms within one to two hours of consuming meals prepared with *Kaempferia galanga* powder (KGP) and presented to the emergency room (ER) within six hours. Symptoms included cardiac arrhythmias, hypotension, perioral paraesthesias, paraesthesias of extremities, dizziness, vomiting, stomach aches, numbness and loss of consciousness. In ON, five severely ill cases were admitted to intensive care. One person initially included in the first ON restaurant case cluster was asymptomatic after the consumption of aconitine-containing foods. The average and median age of BC and ON symptomatic cases was 58 yrs (35 to 84), 55.5% were male.

### 2.2. Blood Aconitine Levels

Blood was collected in 9 of 11 ON cases who consumed the restaurant meal. Aconitine was detected in seven cases and one asymptomatic male who did not meet the case definition. Inconclusive results were obtained from the blood of one symptomatic female who reported less severe symptoms.

### 2.3. Aconitine Testing

The results of aconitine toxin testing are described in Table 1 (BC) and Table 2 (ON).

KGP collected in BC (Brand A) was tested for aconite alkaloids by two laboratories. Both labs detected aconitine alkaloids, as shown in Table 1, in the leftover KGP brought to the ER by those involved in the cases, who used it to prepare a chicken dish (1036 ppm and 5900 ppm). Aconitine was also detected in 1 of 11 unopened packages collected from a retail store (Brand A, 6100 ppm), with lower levels found in the remaining 10 unopened samples (0.013 ppm to 0.030 ppm). Case and non-case 70 g packages were portioned from one larger bag in-store. Two other KGP brands collected at retail as control samples (B, C) were negative for aconitine.

KGP collected in ON (Brand D, lot X, Y, Z) was retrieved from the restaurant and distributor. Aconitine was detected in one of two samples collected from the restaurant (lot X, 5500 ppm; toxin absent in lot Y). An additional 10 samples from an unopened box collected from the distributor containing 57 unopened packages were also tested in combination (lot Y and lot Z), and aconitine was not detected in these samples.

### 2.4. Plant DNA Testing

*Aconitum* spp. was detected in 5 of 15 BC samples (Brand A), which targeted three areas common to plant DNA using DNA barcoding methods, described in Table 3. Follow-up testing specific to toxic plants of the *Aconitum* genus showed 14 of 15 samples were positive for *Aconitum* spp. The control brand samples tested negative for *Aconitum* spp.; one tested positive for *Cuminum* spp., not *Kaempferia* spp. (Brand B), and the second had no detectable plant DNA present (Brand C). A wide variety of other plants were also detected in all samples, including *Oryzae* spp. (rice), cumin, dandelion, dill, fennel, avocado and other shrubs.

In ON samples, *Aconitum* spp. was detected in the implicated opened KGP from the restaurant (Brand D, lot X) and no other samples (Brand D, lot Y and lot Z). *Kaempferia elegans* (peacock ginger), fennel, dill and cumin were found in lot Y, and cumin was found in lot Z.

### 2.5. Investigation Comparisons

#### 2.5.1. Outbreak Identification

In BC and ON, ER physicians were the first point of contact for ill cases, for which they assessed and managed patient illness. Poison Control Centres (PCCs) were consulted for advice on patient management for suspected toxin illness based on case symptoms, onset and food history. PCCs alerted on-call Medical Health Officers (MHOs) at regional public health (PH) offices (in BC). In ON, provincial police were the first to contact regional PH offices, followed by hospitals. PH offices in both provinces deployed inspectors to initiate case interviews and conduct on-site investigations. ER physicians’ communications to their network included the ON provincial Ministry of Health.

#### 2.5.2. Premises Inspections and Investigations of Product Distribution

A timeline of outbreak events is shown in Figure 1 (BC) and Figure 2 (ON).

In BC, KGP products sold at the store were removed from sale by PH inspectors 13 days after cases illness and exposure. PCC and PH inspectors requested assistance with coordinating laboratory testing of the implicated KGP and information gathering meetings. Factors resulting in delays in BC included identifying the store, receiving implicated KGP from the ER, finding a lab provider to test for aconitine and obtaining standards for the test. Case tracing was hampered because the phone number from ER records was incorrect, and cases could not recall where the spice was purchased in a mall containing many similar businesses. A delay sending the implicated leftover KGP to PCC, combined with communication delays requiring translation services, resulted in delayed inspection.

Once the product location and a photo of the leftover KGP were available, PH inspected the store and held the product. An investigator fluent in Chinese was required to attend the store inspection, interact with the store owner and assist in identifying products with Chinese labels. On-site investigations at the store revealed the owner bulk-purchased two 454 g (1 lb) bags of KGP from an ON distributor, which were received on 4 November 2021. The owner then repackaged the spice into smaller individual packages of 70 g. Repacking was carried out using a weight scale and scoop, and individual packages were sold in store to the public. Eleven packages were ordered and held in the store, with all prior product having been sold and distributed. An invoice linked to this product identified a single purchase in January 2020 from one ON importer who sourced the product from China.

Because the KGP was sold from stores specializing in TCM and herbs, it was uncertain whether the KGP was marketed and sold as a natural health product (NHP) or as a food spice. This was an important distinction to federal authorities as jurisdiction over NHPs resided with two different agencies. Requests to both agencies were initiated; however, toxin testing was not available at the time of request. Aconitine toxin was tested in a local laboratory with NHP expertise; however, a further delay occurred in procuring aconitine standards for the test.

Traceback of the ON-based importer revealed that a single shipment of 250 kg of KGP was received from China two years previously but the importer could not be located. The BC store owner sold imported products to 19 retailers. At the time of this investigation, no product from this shipment could be located at other retailers. Follow-up was hampered by poor record-keeping at the retail store, and available records when reviewed indicated the store owner may have received KGP from other importers. Once a positive identification of aconitine was received, a health advisory notification to the general public (PHN) occurred. Because cross-contamination during repackaging in the store could not be ruled out, and as no other product was available on the market, no further actions were taken.

In ON, the restaurant was closed by PH inspectors within 11 h of cases attending the ER. Food history interviews with family members of cases were integral in the identification of KGP as the exposure source. PH inspectors followed up with the product distributor discovering that KGP was distributed in two other ON cities. Interviews with the distributor at the warehouse were conducted in Mandarin due to language difficulties and invoices were in Chinese, requiring translation. The distributor indicated that the product had not been recently imported, and they did not compare shipping manifests to invoices or products ordered. Invoicing of products received on the same day had the same product shipment code, suggesting mislabeling of *Aconitum* as KGP. PH inspectors conducted on-site inspections to verify recall effectiveness within 19 days, as well as to ensure the product was removed from the shelves of stores and not available for sale. In total, 17 grocery stores were inspected and 22 bags of product seized or discarded among the three cities.

Three different lot codes of KGP were tested. A high level of aconitine (5500 ppm) was detected in the open restaurant sample of KGP that shared the same shipment lot code identifier as the imported *Aconitum* natural health product powder (“Radix Aconti Kusnezoffi”). A Health Risk 1 designation supported a Class 1 Food Recall Warning. Investigations estimated the distributor mislabeled 28.1 kg of product (or 61 packages of 454 g). This product was imported in January of 2020 and distributed to four provinces: BC, ON, Alberta (AB) and Quebec (QC). Federal investigators conducted a total of 26 recall effectiveness checks in all provinces, finding no affected product on store shelves. Focused searches of importers of KGP and *Aconitum* powders included store visits to assess handling of food and NHP products. Recall effectiveness checks were repeated in ON stores again in October following the new case report, but no product was found during any on-site visit.

#### 2.5.3. Communication Strategies and Interactions Among Agencies

In BC, initial communications were facilitated with provincial assistance to coordinate information between PH offices and other partners for laboratory testing support. Following the positive aconitine test result from the research lab, federal partners were contacted with the aim of sharing and reviewing the testing methodology, and to provide foreknowledge of follow-up actions, including the PHN issued that day [17]. Seized packages of KGP were transferred to federal agencies, who provided confirmatory testing using plant DNA methods, and once developed, for aconitine alkaloid. The PHN was translated into simplified and traditional Chinese and several TV and newspaper interviews conducted.

In ON, initial communications were facilitated by the hospital and ER physicians, who contacted the PCCs for a toxicological consultation. PCCs advised that the symptomology aligned with aconitine poisoning and advised on treatment options. Physicians communicated within their network to notify others of this potential mass poisoning event. This outreach alerted physicians in provincial agencies, who also contacted PH offices. Provincial investigators worked with PH inspectors in identifying hospitalized cases of individuals who ate at the restaurant, and information-sharing calls determined that an Outbreak Investigation Coordination Committee (OICC) should be formed. This committee was led by federal partners, enabling information sharing among all provinces/territories and the coordination of roles during this outbreak.

Multiple PHNs and advisories to secure national health networks (PHAs) were issued by PH, provincial and federal partners. Two PHAs were posted on Canada’s national public health intelligence site (CNPHI) two days after illnesses occurred, to provide information to all stakeholders about the ongoing investigation, including that a national OICC had been activated. The province issued a memo, linked to the food recall warning, requesting physicians inquire about KGP spice consumption should symptoms occur in new cases, which was also shared with communities’ clinicians, ERs and national OICC partners to align messaging in all jurisdictions. Federal authorities provided the Canadian Surveillance System for Poison Information (CSSPI) and Toxicovigilance Canada information for distribution to poison centres across Canada and notified the FAO/WHO International Food Safety Authorities Network (INFOSAN) regarding the recall notice, prompting Hong Kong authorities to follow up with the exporting company.

Regional PH agencies issued a PHN to the community in their jurisdiction warning residents not to consume KGP 3 days after the illnesses occurred. The PHN informed residents and operators (in various languages) that there was a hazardous KGP product that should be discarded if present on their shelves at home or being used or sold in their premises. The PHN was updated the following day to include the food recall warning for the Brand X KGP product [18,19]. Federal agencies issued a PHN and social media messaging 5 days after the illnesses, informing people in Canada of the illnesses linked to the food recall [20].

## 3. Discussion

In BC and ON investigations, high levels of aconitine alkaloids were found in the KGP spice powders, confirmed by the presence of *Aconitum* plant DNA in the samples. In one study, processed *Aconitum* roots were reported to contain between 0.0041 and 0.0061% of aconitine, and crude roots between 0.14 and 0.19% of aconitine [4]. In these poisonings, levels in excess of 6000 ppm (i.e., 0.6%) were high enough to elicit severe symptoms requiring hospitalization. *Aconitum* plant root powders look similar to *Kaempferia galanga* root powders, making misidentification possible. One review of poisonings over a seven-year span found *Aconitum* roots likely contaminated multiple commonly used herbal remedies, which were linked to 43 cases [12]. The review found that the identification of roots, once ground into fine pieces, would be difficult for experts [12]. In our investigations, mislabeling at the ON distributor occurred as the same shipment lot codes were used to identify both KGP and *Aconitum* products, although the product codes differed. Because both products were ground, the identification of the plant material using shipment codes likely contributed to misidentification. Traceback to the importer in both provinces implicated products imported in January 2020 from Hong Kong (“*Kaempferia galanga* Powder” and “Radix Aconiti Kusnezoffii Powder”), which were labeled with the same product name, “*Kaempferia galanga* Powder”. The manufacturer in China had previous labeling issues. It is plausible this was the same product in both provinces, although this could not be confirmed. Import searches identified five other BC companies that may have imported KGP and *Aconitum* to Canada from the specific Hong Kong manufacturer. The findings were that *Aconitum* is typically imported as a root or slice, not as powder and not sold as a food item. Where *Aconitum* was sold as an NHP, clear instructions were provided that the product should be boiled for at least two hours prior to consumption, and that a minimum of one hour was required to reduce toxicity [5]. Preparations must be properly prepared and dosed to ensure safe use. In Canada, *Aconitum* roots used in NHPs must be processed before use to contain no more than 0.02% of alkaloids, and a maximum dose of 30 mg/day for the root must not be exceeded [21]. Notably, outbreaks have occurred when large quantities of roots are used in herbal soups as medicinal foods, even after boiling for 24 h [8].

In ON, there was a wider distribution of this product than in BC, leading to an increased risk to the public. Of concern with this product were the early import date, large distribution and likelihood the product could still be in consumers’ homes. Communications were translated into several languages with the aim to reach those in the community who may have previously purchased this product and advise them to discard it. An additional poisoning did occur in October 2022 in a single case of someone who had purchased the product before the recall occurred. The case reported hearing about the recall but did not check the lot code of the product before using it at home. This case recovered after a serious illness requiring intubation and hospitalization. This prompted a new PHN and repeat of product recall activities, but the implicated KGP was not found for sale [22].

Similarities in actions and differing challenges were experienced by BC and ON during aconitine poisonings in 2022. Symptoms of heart palpitations, flushing and irregular breathing prompted cases to visit emergency rooms within six hours of consuming meals. ER physicians in both provinces involved PCCs who diagnosed aconitine rapidly based on symptoms. In BC, PCCs contacted public health offices directly, while in ON, ER staff alerted the police who then contacted public health offices. In both provinces, public health actions to hold the product (BC) and close the implicated premises (ON) occurred once premises were identified; however, in BC, multiple issues caused inspection and testing delays, as described in Table 4 and Figure 1.

Federal agency involvement was delayed in BC as jurisdiction over KGP was unclear and toxin identification methods were unavailable. As the store sold both categories of products, this created uncertainty during the initial review. Plant DNA testing was initially offered, but the information conveyed was that the treatment of spices, such as boiling, high-heat drying and grinding, could lead to sub-optimal testing results, described as a ‘long shot’. A decision was made to seek a laboratory that could test for the food toxin, rather than test for the plant material contained in the spice. Aconitine was detected in a NHP lab 27 days after the illnesses occurred due to delays in obtaining reference standards for the method. Federal authorities deployed the available plant DNA testing methods, which found *Aconitum* spp. in sand ginger spice 53 days after illnesses, and developed aconitine methods to confirm the initial findings of aconitine 68 days after illnesses.

In ON, minimal delays occurred throughout the investigation. Aconitine toxic alkaloid testing was immediately available through federal laboratories, as a methodology was in place following the BC experience. This was the first mass toxin event in the history of Ontario’s foodborne illness outbreaks. The PCC was integral in informing a treatment plan for cases in the ER, which led to the policy decision to revise the ON foodborne illness outbreak response protocol (FIORP), an action also taken in BC. In both provinces, updates included diversified communications and roles with poison control partners, along with the inclusion of food poisoning toxin investigations into the respective provincial FIORPs, and furthered relationship building [23,24].

Plant DNA testing provided additional insight into spice contents. The powders contained *Aconitum* plant material as well as 15 other plant species: rice, cumin, dill, fennel, avocado, dandelion and vines from peony, kiwi and other shrubs. Of the 20 samples tested, 45% contained various rice species and 35% contained other taxa. TCM formulations commonly contain many compounds, and the variety of plants found in the mislabeled spice was expected. However, during the BC investigation, the leading hypothesis was that cross-contamination during packaging occurred in the store. In retrospect, the absence of *Kaempferia* and presence of *Aconitum* along with other taxa should have prompted further introspection as to the cause.

The absence of *Kaempferia galanga* in the control and unimplicated lots is suggestive of food misrepresentation and adulteration. Of note, none of the 20 samples, including the KGP control spices collected from retail (Brands B and C) or the lots not mislabeled in ON (lots Y and Z, Brand D) contained *Kaempferia galanga* (lot Y from ON contained *K. elegans*). It is possible that *Kaempferia galanga* was present; however, it may have been in quantities too small to detect using these methods [25]. Substitution is common in spices [26]. In 2022, China was the second-largest exporter of spices in the world, with exports valued at USD 514 million and ginger products accounting for the majority (87%) of this value [27]. In Canada, adulteration, addition of undeclared substances, substitution and misrepresentation are defined as illegal activities [28]. Although food identification detection methods are currently expensive and require standardization, a combination of traditional, chemical and molecular methods is recommended to detect biological adulterants, such as aconitine [25,28,29,30]. Intentional adulteration of unapproved non-biological substances, such as sand, marble and floral parts, has been demonstrated in spices [30] and may be why no plant DNA was detected in one control.

## 4. Conclusions

Food poisoning outbreaks linked to toxins were unusual events in both provinces, requiring coordination and collaboration with new partners. Neither province was initially aware of Toxicovigilance Canada, a network of poison specialists available for consultation on poisoning events, including toxicology laboratory response information. Contacting experts within this network during investigations would improve access to laboratory test providers and toxicology experts on rare poisonings. The rapid testing response time in ON for aconitine alkaloids was possible because methods were developed following the BC event. Blood specimens were successfully tested for aconitine alkaloids. However, as blood alkaloids may dissipate within 24 h in blood, urine samples are recommended [4]. No calculations of the ingested toxic dose could be calculated because case weights, amounts of contaminated meals ingested or amounts of spice used to prepare meals were not collected in either province. Plant DNA testing confirmed the aconitine test results and provided valuable information about the spice contents.

Agencies in both provinces were fortunate to have frontline staff available who were fluent in Mandarin and Cantonese, which is essential during investigations for communications with operators and cases and to identify mislabeled products, read invoices and product labels and create news releases and educational notices for public distribution. The range of KGP distribution across Canada, coupled with the long shelf life of spices, creates concern that additional cases may occur in consumers who purchased spice before the recall and did not hear or act on the public messaging to discard it. This did occur in ON in a single case two months following the original poisonings in the restaurant.

In BC, a more patient-centric database shared between health authorities, hospitals and other provincial agencies is required. Exchange of information still largely occurs through emails, and incorrect and incomplete information led to delays in contacting cases and follow-up. Initial follow-up by federal investigators was limited as operationally they have a higher threshold before precautionary measures and investigational activities are undertaken. In BC, because the original KGP spice packet was opened, cross-contamination during in-store packaging could not be ruled out. This hampered formal investigations into distribution until independent, confirmatory testing by federal authorities on unopened KGP packets was undertaken. Poor record-keeping coupled with the complications of the use of Chinese languages on invoices meant investigators were not led to the mislabeling issue until the ON outbreak occurred six months later.

## 5. Materials and Methods

Partner roles described here are named generically. Names of agencies and their roles in the investigation are provided in Appendix A, along with acronyms. Abbreviated summaries of BC and ON investigations, provided in synopses in Appendix A, are provided for context. Investigations were compared in joint meetings in both provinces to review the findings and highlight issues encountered.

### 5.1. Case Definition for Aconitine Poisoning

Cases were defined as residents of BC and ON with exposure to foods containing aconitine with symptoms and onsets consistent with aconitine poisoning. More specifically, cases were defined as having consumed KGP in meals at an ON restaurant (on 27 or 28 August 2022), or in home-made meals made with KGP purchased at a store in BC (on 4 February 2022) or from an ON store with an identical lot code linked to the recalled KGP; AND with cases experiencing, within two hours of consuming aconitine-contaminated foods, any of the following: heart arrhythmia, ataxia, dizziness, numbness/tingling, paresthesias, decreased level of consciousness, with or without diarrhoea, stomach cramps, nausea, vomiting, fever or headache.

### 5.2. Aconitine Test Methods

In these investigations, four laboratories performed three types of testing: food samples were tested for *Aconitum* alkaloids in two laboratories, food samples were tested for *Aconitum* DNA and clinical blood specimens were tested for toxic alkaloids. In BC, methods were developed for aconitine in a research laboratory based on referenced methods used for dietary supplements and botanical materials. Following this test, the presence of *Aconitum* DNA was analyzed in a federal laboratory. The aconitine testing method used by the research laboratory was shared with a second federal laboratory. Federal laboratories tested all retrieved implicated food samples from all investigations, and in ON, blood specimens were tested for alkaloids in a forensic laboratory. Summarized descriptions of the methods are found in Table 5, with details described in the cited references.

## Figures and Tables

**Figure 1 toxins-17-00125-f001:**
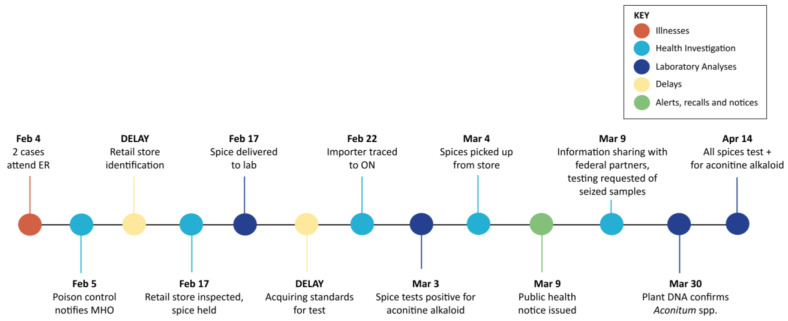
Outbreak investigation timeline in BC, February 2022. Figure abbreviations: ER—emergency room; MHO—Medical Health Officer; ON—Ontario.

**Figure 2 toxins-17-00125-f002:**
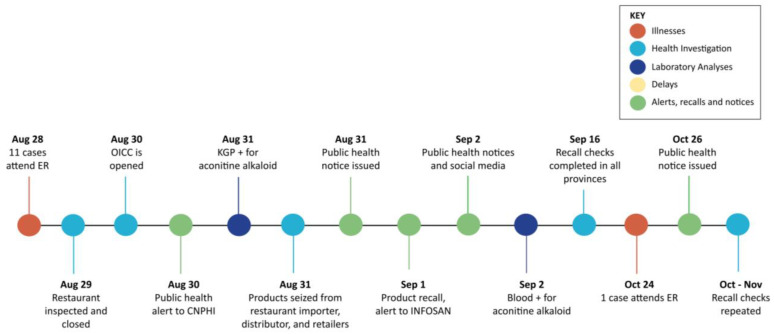
Outbreak investigation timeline in ON, August to October 2022. Figure abbreviations: ER—emergency room; OICC—Outbreak Investigation Coordination Committee; CNPHI—Canadian Network of Public Health Intelligence; KGP—*Kaempferia galanga* powder; INFOSAN—Food and Agriculture Organization (FAO)/World Health Organization (WHO) International Food Safety Authorities Network.

**Table 1 toxins-17-00125-t001:** Results of aconitine testing in BC.

	Case-Implicated Samples		Non-Case Samples			Controls	
Collection site	Emergency room		Retail		Customer	Retail	Retail
Brand/Lot	A/none		A/none		A/none	B	C
Number of samples	1		1	10	1	1	1
Toxin values (ppm) ^a^							
Aconitine	1034 ^b^	5900	6100	0.013–0.030	<0.005	<0.005	0.01
Hypaconitine	17 ^b^	<0.005	<0.005	<0.005	<0.005	<0.005	<0.005
Mesaconitine	NA ^b,c^	<0.005	<0.005	<0.005	<0.005	<0.005	<0.005
Package size (g)	70		70	70	1000	55	55
Open or closed samples	Open		Closed	Closed	Open	Closed	Closed

^a^ Limit of detection < 0.005 ppm; ^b^ tested in research lab, or otherwise, tested by federal toxin lab; ^c^ standard for test not available.

**Table 2 toxins-17-00125-t002:** Results of aconitine testing in ON.

	Case-Implicated Samples		Non-Case Samples		
Collection site	Restaurant		Distributor		
Brand/Lot	D/X	D/Y	D/Z	D/Y	D/Z
Number of samples	1	1	1 ^b^	1 ^b^	1 ^c^
Toxin values (ppm) ^a^					
Aconitine	5500	<0.005	<0.005	<0.005	<0.005
Hypaconitine	<0.005	<0.005	<0.005	<0.005	<0.005
Mesaconitine	<0.005	<0.005	<0.005	<0.005	<0.005
Package size (g)	454		70	70	1000
Open or closed samples	Open		Closed	Closed	Open

^a^ Limit of detection < 0.005 ppm; ^b^ composite of 4 sub-samples; ^c^ composite of 2 sub-samples.

**Table 3 toxins-17-00125-t003:** Plant DNA barcoding identification.

Number of Samples (*n*)	Plant ITS Genus Species (%)	Plant psbA	Plant rbcL	CT Value	Common Name of Plants Identified
BC case sample (Brand A, *n* = 1)	*Aconitum karakolicum* (100.00%)	*A. stylosum* (97.72%)	*A. kusnezoffii* (99.19%)	13.34	Monkshood (*Aconitum*)
ON case sample (Brand D, Lot X, *n* = 1)	*A. talassicum* (97.62%)	*A. stylosum* (95.95%)	*A. kusnezoffii* (100.00%)	14.38	Monkshood (*Aconitum*)
BC sample collected from customer (Brand A, *n* = 1)	*Paeonia lactiflora* (100%)	NA ^1^	*Litsea cubeba* (100%)	29.68	Chinese peony (*P.l*), Chinese evergreen tree in family Lauraceae (*L.c*)
BC sample collected from retail (Brand A, *n* = 1)	*A. karakolicum* (100.00%)	*A*. *stylosum* (98.38%)	*A. kusnezoffii* (99.68%)	13.23	Monkshood (*Aconitum*)
BC sample collected from retail (Brand A, *n* = 1)	*Taraxacum officinale* (98.06%)	*Oryza eichingeri* (99.64%)	*Oryza punctata* (99.03%)	30.11	Common dandelion (*T.o.*), wild rice (*O.e*), red rice (*O p.*)
BC sample collected from retail (Brand A, *n* = 1)	NA	*Oryza rufipogon* (99.10%)	*Oryza sativa* (98.75%)	28.94	Brownbeard rice (O.r.), rice (O.s.)
BC sample collected from retail (Brand A, *n* = 1)	*A. karakolicum* (100.00%)	*A. stylosum (96.19%*)	*Oryza punctata* (98.55%)	22.73	Monkshood, red rice
BC sample collected from retail (Brand A, *n* = 1)	*A. karakolicum* (99.68%)	*A. stylosum* (98.11%)	*Oryza sativa* (98.75%)	21.73	Monkshood, rice
BC sample collected from retail (Brand A, *n* = 3)	NA	NA	*Oryza glaberrima* (99.52%, *n*=2, 99.31%)	29.69, 28.73, 25.51	African rice (*O.g.*)
BC sample collected from retail (Brand A, *n* = 1)	NA	*Persea americana* (98.10%)	*Oryza punctata* (99.36%)	30.04	Avocado (*P.a.*), red rice
BC sample collected from retail (Brand A, *n* = 1)	NA	*A. flavum* (97.81%)	*Oryza punctata* (99.52%)	22.22	Monkshood, red rice
BC sample collected from retail (Brand A, *n* = 1)	NA	*Actinidia chinensis* (98.41%)	*Machilus japonica* (93.48%)	30.62	Golden kiwifruit vine (*A.c.*), Chinese flowering vine (*M.j.*)
ON sample collected from restaurant (open Brand D, Lot Y, *n* = 1)	*Anethum graveolens* (98.07%)	*Anethum foeniculum* (99.13%)	*Kaempferia elegans* (95.16%)	NA	Dill (*A.g.*), fennel (*A.f.*), peacock ginger (*K.e.*)
ON sample collected from retail (closed Brand D, Lot Y, *n* = 1)	*Anethum graveolens* (97.15%)	*Anethum foeniculum* (99.02%)	NA	NA	Dill, fennel
ON samples collected from retail (closed Brand D, Lot Z, *n* = 2)	*Cuminum cyminum* (87.30%)	NA	NA	NA	Cumin
Control sample of sand ginger (Brand B, *n* = 1)	*Cuminum cyminum* (100%)	*Cuminum cyminum* (99.65%)	*Cuminum cyminum* (99.68%)	31.13	Cumin
Control sample of sand ginger (Brand C, *n* = 1)	NA	NA	NA	NA	

^1^ NA—no activity from target, *Aconitum* spp. and other plant spp. absent.

**Table 4 toxins-17-00125-t004:** Investigational delays from time of illness.

Delays from Time of Illness	BC	ON
to site visit and product holds	13 days	11 h
to follow-up at retail and distributor levels	13 days	14.5 h
to importer traceback	18 days	1.5 days
to confirmatory toxin testing of case blood	ND	5 days (RD + 7 days)
to confirmatory toxin testing of spice	27 days	3 days
to confirmatory *Aconitum* plant DNA	53 days (RD + 12 days)	RP
to public health notice	32 days	3 days

ND—not done; RD—reporting delay time; RP—test conducted for research purposes.

**Table 5 toxins-17-00125-t005:** Aconitine and *Aconitum* detection methods.

Laboratory (Location)	Methods
BC Institute of Technology Natural Health and Food Products Research Group (Burnaby, BC)	KGP was tested for toxic alkaloids via LC/MS methods based on the AOAC International Official Method of Analysis for the determination of *Aconitum* alkaloids aconitine, mesaconitine and hypaconitine in dietary supplements and botanical materials [31,32]. Chromatographic separation was achieved on an Agilent 1290 Infinity II LC system coupled with an Agilent 6420 Triple Quad Spectrometer (Agilent Technologies Inc., Mississauga, ON, Canada) using a Waters Xterra RP18 column, 2.1 × 50 mm, 3.5 μm (Waters Ltd., Mississauga, ON, Canada).
Canadian Food Inspection Agency (federal) Toxin Laboratory (Saskatoon, Saskatchewan)	A representative sub-sample of KGP was extracted with an acidified mixture of methanol and water for 60 min at 60 °C while shaking. Extracts were cooled, centrifuged, diluted and analysed using a tandem mass spectrometer coupled with liquid chromatography (LC-MS/MS) (Sciex 5500 QTRAP, Framingham, MA, zUSA) [33]. Chromatographic separation was achieved using an Acquity BEH C18 column, 2.1 × 30 mm, 1.7 µm (Waters Ltd., Mississauga, ON, Canada). Selected reaction monitoring (SRM) transitions were acquired following positive-ionization electrospray: 646.3 > 586.3, 646.3 > 526.3, 646.3 > 368.3.
Canadian Food Inspection Agency (federal) Plant DNA Laboratory (Ottawa, ON)	To test KGP for *Aconitum* DNA, samples were assayed by DNA barcoding [34,35] at the internal transcribed spacer (ITS), the photosystem II protein D1 (psbA) and ribulose-1,5-bisphosphate carboxylase/oxygenase large subunit (rbcL) loci to detect the presence of *Aconitum* plant DNA using bidirectional Sanger sequencing with a BigDye Terminator v3.1 Cycle Sequencing Kit (Life Technologies Inc., Carlsbad, CA, USA) on an ABI 3730xl Genetic Analyzer (Life Technologies Inc., Carlsbad, CA, USA). An assay targeting a region of the ITS locus specific to toxic plants of the *Aconitum* genus [36] was also adapted to a TaqMan Real-time PCR assay (qPCR) and ran on a ViiA7 instrument (Life Technologies Inc., Carlsbad, CA, USA), using *A. napellus* as a positive control. TaqMan qPCR primers and probe sequences were the following (5′ 3′): F-ACGGTCGGCACAAATGTT, R-CGACGCGTCTTGATGTCTTT, PRB-FAM/CGGTCAGTG/ZEN/GTGGTTGTATTTCTCATCC.
Centre of Forensic Sciences (Toronto, ON)	Clinical blood samples were tested for toxic alkaloids. Submitted blood samples underwent sample preparation and were analysed using a Liquid Chromatography Quadrupole Time-of-Flight Mass Spectrometry (LC-QTOF-MS) method to screen for various over-the-counter, recreational and prescription drugs, poisons and metabolites, including acotinine. A Waters Acquity UPLC I-Class System, with an Acquity UPLC HSS C18 1.8 µm 2.1 × 150 mm column, was used for chromatographic separation coupled with a Xevo G2-S QToF for identification. Aconitine was reported as detected once the minimum acceptance criteria for mass error (ppm), retention time error and expected fragment counts were met.

## Data Availability

The original contributions presented in this study are included in the article/Appendix A. Further inquiries can be directed to the corresponding authors.

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
