# Peer review of "Learnings from Separate Aconitum Poisonings in British Columbia and Ontario, Canada in 2022"

_toxins, 2025, doi:10.3390/toxins17030125_

Round 1
Reviewer 1 Report
Comments and Suggestions for Authors
The manuscript gives a comprehensive study into the outbursts of aconitine poisoning incidents that occurred in British Columbia and Ontario, Canada. It presents well-structured epidemiological data, analytical methods, and public health responses, contributing valuable insights on handling of foodborne toxicological events.
Despite the strengths of the manuscript, there are some areas to revise.
I could not get access to full-length article #7, what are Chuanwu and Fuzi?
I could not find the reference #13, if no relevant reference is found it is better to remove it.
Minor points
Reference #4 has doi: 10.1080/15563650902904407
Reference #6 has a title “Aconite poisoning over 5 years: a case series in Hong Kong and lessons towards herbal safety”
Reference #8 has doi:10.3390/toxins6092605.
Reference #16, page numbers are 569-573.
Page 3, line 84: Use non-abbreviated word for ER.
Page 6, Figure 1: I could not find definition for MHO, the abbreviations would be good to be explained in the figure legend.
Page 8, Figure 2: the abbreviations would be good to be explained in the figure legend.
Author Response
Thank you very much for taking the time to review this manuscript. Please find the detailed responses below and the corresponding revisions/corrections highlighted/in track changes in the re-submitted files.
No questions for general evaluation were noted, thank-you.
Comments 1: I could not get access to full-length article #7, what are Chuanwu and Fuzi?
Response 1:
Article #7 by So et al (2020) is from a Sage publication. The full URL is https://journals.sagepub.com/doi/10.1177/102490790000700406?icid=int.sj-abstract.similar-articles.6 and trust this link will assist you to access this article.
Fuzi and Chuanwu are Mandarin words (pinyin) for the species of aconite plants native to China. The meaning of the words are linked to the dried aconite root ingredient that is used in Traditional Chinese Medicine. In one review article previously cited (article #4), Chan (2009) describes these as
“Chuanwu” (the root tuber of A. carmichaeli), “caowu” (the root tuber of A. kusnezoffii), and “fuzi” (the lateral root tuber of A. carmichaeli)
We did not include this information into the introduction, however, in many Asian articles, these are the names referenced in addition to latin botanical names for roots of Aconitum in herbal medicines.
For clarity, we added this to the introduction to make clear the origins of these names on line 49 and 50 in the introduction, and included article #4 in the citation list for this change. The text included is “In TCM and for homeopathic preparations of herbal soups and meals, Aconitum roots (named “fuzi”, “caowu”, and “chuanwu”) and leaves are processed…”
Comments 2: I could not find the reference #13, if no relevant reference is found it is better to remove it.
Response 2: Article #13 is an online posting of illnesses reported by a local health unit and may be found at this link, although it does not show up in the referencing format for the Toxins journal.
https://www.sfmms.org/news-events/sfmms-blog/sfph-health-alert-two-cases-of-accidental-aconite-poisoning-from-medicinal-herbs-purchased-in-san-francisco-chinatown-shop.aspx
I do agree this is a poor reference link. This addition was made to demonstrate the rarity of illnesses in North America. We have, accordingly, revised the citation to include the updated URL on page 450 and 452 of the manuscript.
Minor Comments 3:
3.1 Reference #4 has doi: 10.1080/15563650902904407
3.2 Reference #6 has a title “Aconite poisoning over 5 years: a case series in Hong Kong and lessons towards herbal safety”
3.3 Reference #8 has doi:10.3390/toxins6092605.
3.4 Reference #16, page numbers are 569-573.
3.5 Page 3, line 84: Use non-abbreviated word for ER.
3.6 Page 6, Figure 1: I could not find definition for MHO, the abbreviations would be good to be explained in the figure legend.
3.7 Page 8, Figure 2: the abbreviations would be good to be explained in the figure legend.
Response 3: Thank you for noticing these errors.
3.1 The DOI is added to reference #4 on line 430.
3.2 The title of reference #6 has been corrected on line 434
3.3 The DOI is added to reference # 8 on line 438-439
3.4 The page numbers have been added to reference #16 on line 457
3.5 The non-abbreviated word has been written out as “emergency room” on line 84
3.6 It is understandable this was not clarified. In our first drafts of the manuscript, prior to submission, we did create a supplemental table of abbreviations, with the aim to generically describe the many roles of regional, provincial and federal partners in the investigation (Supplemental Table 1). Our local abbreviations for commonly used words are so ingrained we do not notice them and appreciate you pointing this out.
On line 153, just below Figure 1, abbreviations have been given as “Figure abbreviations: ER-emergency room; MHO-Medical Health Officer; ON-Ontario”.
In addition, on line 136, after the description of Medical Health Officer, the abbreviation is provided in brackets.
3.7 Similar to Figure 1, abbreviations have been clarified on line 194 to 196 as “Figure abbreviations: ER-emergency room; OICC-Outbreak Investigation Coordination Committee; CNPHI-Canadian Network for Public Health Intelligence; KGP-Kaempferia galanga powder; INFOSAN-Food and Agriculture Organization (FAO)/World Health Organization (WHO) Inter-national Food Safety Authorities Network.”
We also added the terms above that were missing from the first submission of Supplemental Table 1.
In addition all other references were checked and found article 25 was also missing the DOI information, this has now been added on line 478.
Reviewer 2 Report
Comments and Suggestions for Authors
This case study descibed a mass toxin poising event caused aconitum. The authors also investigated throughly the sand ginger spice in Ontario and Britain Columbia. This work is evidentally solid and of great significance to public health, and I suggest this manuscript to be accepted after minor revision.
1. Line 100, was this postive sample the same lot with the other 10 samples?
2. Sector 5.2 & Table 5, I suggest that the authors put in the exact parameters for the quantitation of aconines, such as MS1/MS2. Also should included were the primers used in qPCR test.
Author Response
Thank you very much for taking the time to review this manuscript. Please find the detailed responses below and the corresponding revisions/corrections highlighted/in track changes in the re-submitted files.
No issues were noted for the general evaluation, thank you.
Comments 1: Line 100, was this postive sample the same lot with the other 10 samples?
Response 1: Yes. Although no lot number was identified on any of the packages, all samples came from one larger 454 gram (1 lb) bag of sand ginger. They were packaged in the shop into 70g portions, described in Section 2.5.2.
I agree this is not clear when reading through the results section. To clarify this additional text is added to lines 103 and 104, Case and non-case 70 gram packages were portioned from one larger bag in-store.” Trust this makes clear that all samples were from one source, without providing the details described in the later investigation sections.
Comments 2 Sector 5.2 & Table 5, I suggest that the authors put in the exact parameters for the quantitation of aconines, such as MS1/MS2. Also should included were the primers used in qPCR test.
Response 2:
We agree that the descriptions of methods in this paper are not fully described here. To clarify in section 2, we have amended the last sentence on line 403-404 to state ” Summarized descriptions of methods are found in Table 5, with details described in cited references.”
In addition, the methods authors have provided additional details to Table 5 on line 406 as requested, as shown in the edited manuscript and described below.
From CFIA Toxin Laboratory this was added: Selected Reaction Monitoring (SRM) transitions were acquired following positive-ionization electrospray: 646.3>586.3, 646.3>526.3, 646.3>368.3.
From CFIA Plant Laboratory this was added: TaqMan qPCR primers and probe sequences were the following (5’ 3’): F- ACGGTCGGCACAAATGTT, R- CGACGCGTCTTGATGTCTTT, PRB- FAM/CGGTCAGTG/ZEN/GTGGTTGTATTTCTCATCC.
From Forensics Laboratory this was added: Aconitine was reported as detected once the minimum acceptance criteria for mass error (ppm), retention time error and expected fragment counts were met.
Reviewer 3 Report
Comments and Suggestions for Authors
This is a well written review of several human Aconitum poisonings. Intoxication was confirmed with both chemical identification of Aconitum alkaloids and DNA figerpinting of plant; material and ingesta. The reported clinical signs, laboratory findings, and histologic lesions were review, but were not evaluated in this work. This manuscript will make an informative contribution to your journal.
Author Response
Thank you very much for taking the time to review this manuscript. We appreciate your comment that this work will make an informative contribution to this journal.